# Breast implant surface topography triggers a chronic-like inflammatory response

Valeriano Vinci[1,2,*], Cristina Belgiovine[1,3,*], Gerardus Janszen[4,*], Benedetta Agnelli[2,*], Luca Pellegrino[2], Francesca Calcaterra[1,5], Assunta Cancellara[1,5], Roberta Ciceri[1,5], Alessandra Benedetti[4], Cindy Cardenas[1], Federico Colombo[1], Domenico Supino[1], Alessia Lozito[2], Edoardo Caimi[2], Marta Monari[1], Francesco Maria Klinger[6], Giovanna Riccipetitoni[3,7], Alessandro Raffaele[7], Patrizia Comoli[7], Paola Allavena[1,2], Domenico Mavilio[1,5], Luca Di Landro[4,*], Marco Klinger[1,5,*], Roberto Rusconi[1,2,*]

**Breast implants are extensively employed for both reconstructive and esthetic purposes. However, the safety of breast implants with textured surfaces has been questioned, owing to a potential correlation with anaplastic large-cell lymphoma and the recurrence of breast cancer. This study investigates the immune response elicited by different prosthetic surfaces, focusing on the comparison between macrotextured and microtextured breast implants. Through the analysis of intraoperatively harvested peri-prosthetic fluids and cell culture experiments on surface replicas, we demonstrate that macrotextured surfaces elicit a more pronounced chronic-like activation of leucocytes and an increased release of inflammatory cytokines, in contrast to microtextured surfaces. In addition, in vitro fluorescent imaging of leucocytes revealed an accumulation of lymphocytes within the cavities of the macrotextured surfaces, indicating that the physical entrapment of these cells may contribute to their activation. These findings suggest that the topography of implant surfaces plays a significant role in promoting a chronic-like inflammatory environment, which could be a contributing factor in the development of lymphomas associated with a wide range of implantable devices.**

## Introduction

Breast cancer remains the most prevalent cancer among women globally, with over 2 million new cases reported in 2020, marking an 11.7% increase from the previous year, as per the Global Cancer Observatory (https://gco.iarc.fr/today/home). In the realm of breast cancer treatment, breast reconstruction has become a vital component, offering not only physical restoration but also significant psychological and esthetic benefits for women undergoing mastectomy. In recent years, postmastectomy breast reconstruction has decisively shifted from autologous procedures to implant-based reconstruction, mostly linked to the advantages in terms of minor complications, faster recovery, and reduced healthcare costs (Davis et al, 2020). Furthermore, breast implants are also extensively used in cosmetic surgical procedures. The initial generation of breast implants featured smooth surfaces. However, to prevent potential risks of implant displacement and rotation, and to lower the incidence of capsular contracture, the development and use of implants bearing different types of textured surfaces have gained widespread acceptance (Barnsley et al, 2006; Maxwell et al, 2014). Manufacturers typically classify breast implants based on the characteristic scale of their textured surface features into three categories: smooth (<10 $\mu$m), microtextured (10–50 $\mu$m), and macrotextured (>50 $\mu$m) (https://www.iso.org/standard/63973.html).

The safety of breast implants, particularly in relation to their macrotextured surface, has been questioned since 1997, after the first reported case of breast implant–associated anaplastic large-cell lymphoma (BIA-ALCL) (Keech & Creech, 1997). BIA-ALCL, a non-Hodgkin lymphoma of T-cell origin, was classified as a hematolymphoid neoplasm by the World Health Organization (WHO) in 2017 (Arber et al, 2016). Despite the challenge of quantifying an accurate risk assessment because of limited global reporting and incomplete sales data, breast implants have been inserted into the list of agents with high priority for evaluation by the International Agency for Research on Cancer (IARC) for inclusion in their monographs on carcinogenic risks to humans (Srinivasa et al, 2017; Clemens et al, 2018; Marra et al, 2020). In response to safety concerns, France has specifically banned macrotextured devices and nearly 40 different

---

[1]IRCCS Humanitas Research Hospital, Rozzano, Italy   [2]Department of Biomedical Sciences, Humanitas University, Pieve Emanuele, Italy   [3]Department of Clinical, Surgical, Diagnostics and Pediatric Sciences, University of Pavia, Pavia, Italy   [4]Department of Aerospace Science and Technology, Politecnico di Milano, Milan, Italy   [5]Department of Medical Biotechnologies and Translational Medicine, University of Milan, Milan, Italy   [6]Department of Health Sciences, University of Milan, Milan, Italy   [7]Fondazione IRCCS Policlinico San Matteo, Pavia, Italy

Correspondence: valeriano.vinci@hunimed.eu; c.belgiovine@smatteo.pv.it; roberto.rusconi@hunimed.eu
*Valeriano Vinci, Cristina Belgiovine, Gerardus Janszen, Benedetta Agnelli, Luca Di Landro, Marco Klinger, and Roberto Rusconi contributed equally to this work

---

 

countries have restricted the use of Allergan Biocell breast implants (Maxwell et al, 2014). Allergan's salt loss manufacturing technique, which creates a notably coarse macrotextured surface, is designed to enhance tissue integration and esthetic outcomes. Yet, despite the extremely common use of textured breast implants, there is very limited knowledge on the correlation between implant surface topography and adverse effects in patients.

Albeit genetic factors may contribute to the onset of this lymphoma (Tevis et al, 2019; Laurent et al, 2020), there is growing evidence that the chronic inflammatory state associated with textured prostheses plays a role in fostering a pro-tumoral environment (Mempin et al, 2021; Mankowski et al, 2022). Recent animal studies involving mice and rabbits have highlighted a direct correlation between the inflammation predominantly mediated by T cells and the roughness of the implant surface (Doloff et al, 2021). The microenvironment of BIA-ALCL has been further associated to an abundance of T helper 17 (Th17) CD4$^+$ cells, which are stimulated by cytokines to enhance the inflammatory response, and T-regulatory (Treg) CD4$^+$ cells, which serve to suppress the immune response. This setting can be described as a pro-inflammatory milieu with chronic T-cell stimulation, as evidenced by CD30 expression (Wolfram et al, 2012). Other proposed etiopathological hypotheses for BIA-ALCL in the context of textured implants (particularly macrotextured) include mechanical degradation because of friction and chronic inflammation induced by bacterial biofilm (Bewtra et al, 2022; Alessandri-Bonetti et al, 2023). Despite these associations, a direct causal link between implant texturization and tumor development in BIA-ALCL has yet to be established.

In this study, our objective was to investigate the differential microbial contamination and immune responses elicited by macrotextured and microtextured breast implant surfaces. We hypothesized that macrotextured surfaces would provoke a more pronounced inflammatory response than microtextured surfaces, potentially playing a role in the pathogenesis of conditions such as BIA-ALCL. To explore this hypothesis, we conducted an extensive analysis of periprosthetic fluids collected from patients implanted with breast implants having different surface textures. Our approach encompassed examining the bacterial load, profiling immune cell populations, and analyzing the inflammatory cytokine landscape. Our findings indicated a chronic-like inflammatory environment associated with macrotextured implants. Furthermore, in vitro experiments performed culturing healthy donor-derived peripheral blood mononuclear cells (PBMCs) on model surfaces mimicking the characteristics of both microtextured and macrotextured implants corroborated our initial observations, reinforcing the conclusion that the texture of breast implant surfaces is a critical factor in modulating the immune response in the periprosthetic environment.

## Results

Periprosthetic fluids collected during the removal of both microtextured and macrotextured breast implants were first analyzed for bacterial contamination through a shotgun metagenomic approach (Fig S1; see the Materials and Methods section). The overall bacterial load in the periprosthetic fluids of our patient pool was generally low. However, the bacterial families identified, which included Actinobacteria, Bacteroides, Firmicutes, and Proteobacteria, demonstrated a similar richness of species across both types of implant surfaces (Fig S1A and B). Predominantly, species from the *Bifidobacterium* genus were identified, which are also found in breast milk (Yan et al, 2021). This may suggest a potential link between the mammary glands and the prosthesis pocket. Notably, in two patients having bilateral prostheses with different surface topographies, a reduction in the number of bacterial species was observed in the macrotextured implant compared with the microtextured one (Fig S1C and D).

We then performed a multiparametric fluorescence-activated cell sorting (FACS) analysis on leukocytes derived from the periprosthetic fluid samples to investigate their composition and characteristics (see the Materials and Methods section). The frequency of classical monocytes, which exhibit an inflammatory phenotype and are characterized by CD14$^+$/CD16$^-$ markers, was found to be decreased in fluids associated with macrotextured implants compared with those with microtextured surfaces (Fig 1A and B). Conversely, in macrotextured implants, we observed an increase, albeit not statistically significant, in macrophages, particularly the CD163$^+$CD206$^+$ subset which is known for its immunosuppressive properties (Fig 1C and D). Further analysis of the immune system components showed an increasing trend in the percentage of eosinophils, neutrophils, natural killer (NK) cells, and CD8 cytotoxic T lymphocytes and a statistically significant increase in T-regulatory (Treg) cells—indicative of an immunosuppressive microenvironment and often found in tumors—among CD45$^+$ cells in the periprosthetic fluids (Fig 1E–L). T-cell profiling revealed notable differences in the maturation of T lymphocytes between macrotextured and microtextured implant groups. Specifically, we observed a decreasing trend in naive CD4$^+$ T cells in the macrotextured group, which was paralleled by an increasing frequency of central memory (CM) and effector memory (EM) CD4$^+$ T cells (Fig 1M–O). Similarly, in CD8$^+$ T cells, we reported a statistically significant contraction in the naive subset (Fig 1Q–S), indicating a potential shift toward a more mature immune response. For T-cell activation analysis, we employed markers such as HLA-DR, CD69, and CD30 (Fig 1P, T, and U–Y), the latter being particularly relevant because of its association with BIA-ALCL (Wolfram et al, 2012). In the periprosthetic fluids collected from patients with macrotextured implants, a statistically significant increase was observed in the frequencies of CD69$^+$ and CD30$^+$ cells among CD4$^+$ T cells, underscoring a heightened activation state (Fig 1P, T, U, W, and X). In two unique cases, who had bilateral implants with different textures, we noted a similar trend in the immune response (Fig 1V and Y).

Motivated by our ex vivo findings, we developed an in vitro platform to investigate the immediate immune response to specific breast implant surface topographies. This approach allowed us to study cellular reactions in a controlled environment, free from the confounding factors present in ex vivo samples, such as genetic backgrounds, patient histories, and variations in the duration of implantation. To mimic real-world conditions, we replicated the exact topographies of commercial microtextured and macrotextured breast implants (Figs 2A–C and S2A–D) using polydimethylsiloxane (PDMS). This material was chosen for its similarity

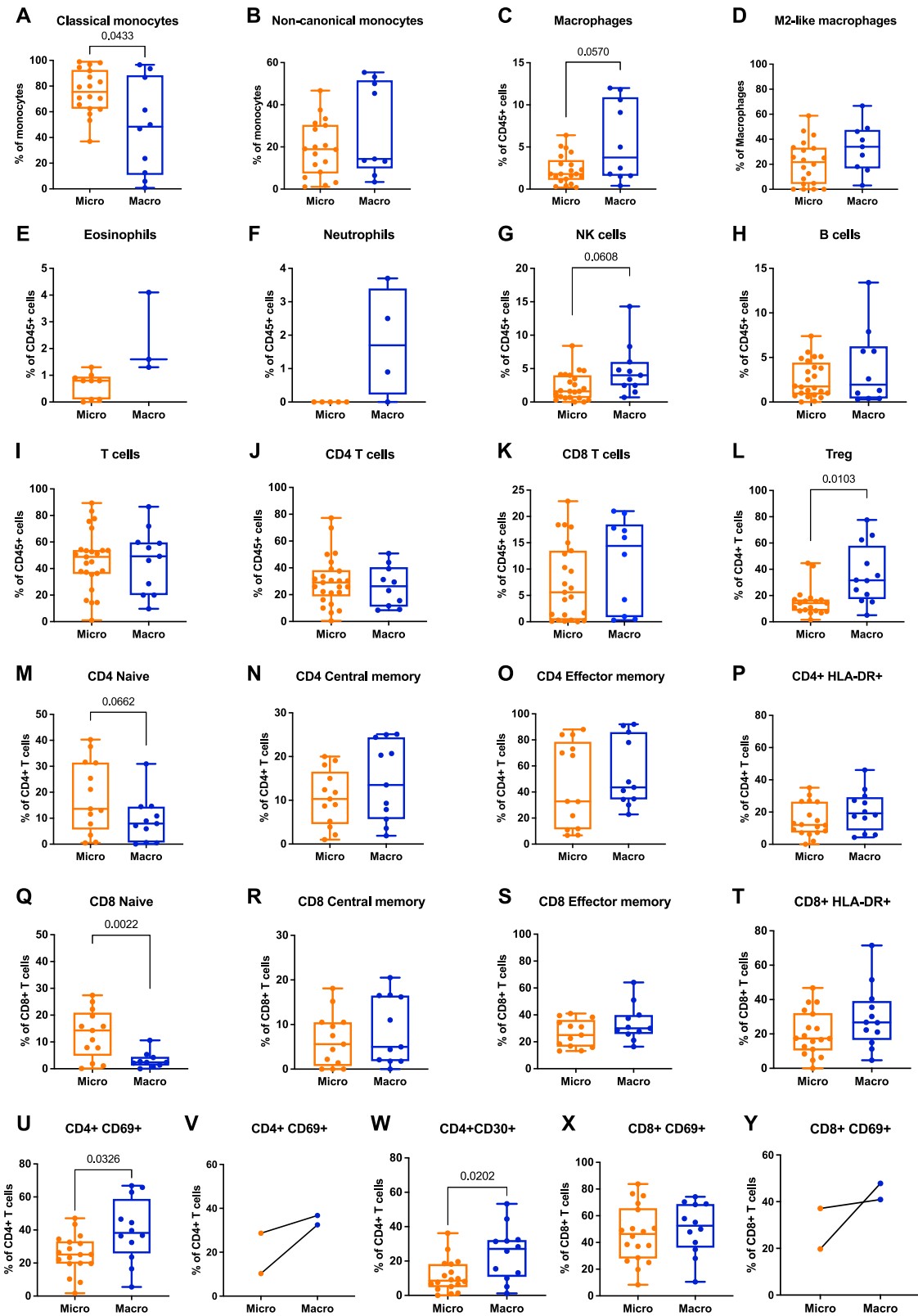

**Figure 1. Flow cytometry analysis of periprosthetic leukocytes from patients with microtextured and macrotextured breast prostheses.**
Frequencies of the different immune subsets calculated as **(A, B)** frequency of viable monocytes, **(C, E, F, G, H, I, J, K)** frequency of viable CD45+ cells, (D) frequency of viable macrophages, **(L, M, N, O, P, U, V, W)** frequency of viable CD4+ T cells, **(Q, R, S, T, X, Y)** frequency of viable CD8+ T cells. Each histogram in the figure represents the

to the silicone used in actual breast implants and for its bio-compatibility, non-toxicity, and ease of casting (see the Materials and Methods section). Human PBMCs from healthy female donors were cultured on microtextured and macrotextured PDMS surfaces for 48 h without any vital stimuli, after which they were analyzed via FACS to assess viability and immune activation (see the Materials and Methods section, Figs 2 and S3). Although the macrotextured surfaces were associated with a marginally higher fraction of ne-crotic cells compared with the microtextured ones, the percentage of viable cells observed in the presence of both microtextured and macrotextured surfaces were similar to those observed in the flat, empty control wells (Fig S3).

No significant differences were observed in the frequencies of monocytes and T cells—including CD4 and CD8 subsets and CD4 Treg—between PBMCs cultured on microtextured and macro-textured surface replicas (Fig 2D–H). Notably, the in vitro model revealed an increase in effector and central memory cells for CD4 (Fig 2J and K) and CD8 (Fig 2N and O) on macrotextured surfaces, in agreement with our ex vivo results. Meanwhile, naive CD4 did not display any variation (Fig 2I), whereas naive CD8 were reduced on macrotextured surfaces (Fig 2M). In addition, activated CD4 and CD8 cells, marked by CD69 expression, were increased in PBMCs cul-tured on macrotextured surfaces (Fig 2L and P). This finding aligns with the ex vivo observations (Fig 1U, V, X, and Y).

To ascertain whether the periprosthetic microenvironment in patients is influenced by the texture of implant surfaces, we conducted an extensive cytokine analysis. This included pro-inflammatory cytokines such as IL6 and IL8, typically elevated in the tumor microenvironment, and TNF-alpha, known for its asso-ciation with T-cell activation. We also analyzed the chemokines CCL2 and CCL5, which are known for their chemoattractant prop-erties for immune cells. Enzyme-linked immunosorbent assay (ELISA) analysis of periprosthetic fluids revealed significantly ele-vated levels of IL6, IL8, and TNF-alpha in macrotextured implants compared with microtextured ones (Fig 3A–C). Although CCL2 levels were comparable between the two groups, a notable reduction in CCL5 was observed in the macrotextured group (Fig 3D and E). These findings, particularly for IL6 and IL8, were further validated using ELLA technology. ELLA's high sensitivity and capacity for simulta-neous multiple immunoassays confirmed these results (Fig S4A and B). In addition, we expanded our analysis using ELLA to include cytokines previously associated with BIA-ALCL, such as IL4, IL10, IL13, IL22, and INFɣ (Turner et al, 2020; Wang et al, 2021; Zhang et al, 2022). This analysis showed significantly higher levels of IL4, IL13, and IL22 in the periprosthetic fluid from the macrotextured group compared with the microtextured group (Fig 3K–O), highlighting a distinct cytokine profile potentially relevant to the pathogenesis of ALCL.

Consistent with these findings, in vitro ELISA analysis revealed that PBMCs cultured on macrotextured model surfaces secreted significantly higher levels of IL6, IL8, and TNF-alpha compared with those on microtextured surfaces (Fig 3F–H). Although the levels of CCL5 remained consistent across both types of surfaces, CCL2 was

found to be lower in the culture supernatants from PBMCs cultured on macrotextured surfaces (Fig 3I and J). These results align with the ELLA findings from culture supernatants, which corroborate the ELISA data for IL6 and IL8 (Fig S4C and D). In addition, further ELLA analysis of cytokines previously linked to ALCL showed that with the exception of IL22, cytokine levels were elevated in supernatants from PBMCs cultured on macrotextured surfaces compared with those on microtextured ones (Fig 3F–H and P–T). This compre-hensive dataset suggests that macrotextured surfaces are more prone to activating immune cell responses, thereby promoting an inflammatory, chronic-like microenvironment.

To elucidate the potential mechanisms underlying leucocyte activation on macrotextured surfaces, we analyzed the distribution of PBMCs, marked with a vital fluorescent label, across different surface textures over time (see the Materials and Methods section). On flat surfaces, PBMCs showed a random distribution (Fig 4A). In contrast, a distinct pattern was observed with microtextured samples (Fig 4B), becoming even more pronounced for macro-textured samples, where PBMCs accumulated within the charac-teristic surface cavities (Fig 4C and D). This behavior suggests a specific interaction of PBMCs with the varying textures, highlighting the role of surface topography in influencing cellular distribution. Distinguishing lymphocytes (red fluorochrome) from monocytes (green fluorochrome) allowed us to further observe distinct be-haviors on these surfaces. Specifically, lymphocytes were seen progressively moving into and accumulating within the cavities of the macrotextured surfaces (Fig 4E). In contrast, monocytes dis-played greater mobility and were predominantly found outside these cavities (Video 1). This behavior was consistent under sterile conditions (Fig 4A and D) and when surfaces were pre-treated with a diluted concentration of *Staphylococcus epidermidis* (Caldara et al, 2022)—a common culprit in breast implant–associated infections—24 h before cell plating (Fig 4E). The unchanged be-havior in the presence of bacteria further emphasizes that the response is driven by surface texture rather than microbial factors. These observations suggest that the activation of lymphocytes on macrotextured surfaces may be significantly influenced by their entrapment within surface cavities. Factors such as the localized concentration of soluble mediators and cell density could be pivotal in driving this activation, highlighting the complex interplay between physical surface features and cellular responses in the immune system.

## Discussion

BIA-ALCL emerges as a complex condition, influenced by genetic predispositions, bacterial biofilms, chronic inflammation, and textured breast implants' properties, indicating a multifaceted pathogenesis (Turner et al, 2020; Wang et al, 2021). In this work, although we did not perform a genetic analysis of our patients'

mean ± SD of the measured parameters. The statistical significance of differences between microtextured and macrotextured prostheses was determined using an unpaired *t* test with Welch's correction.

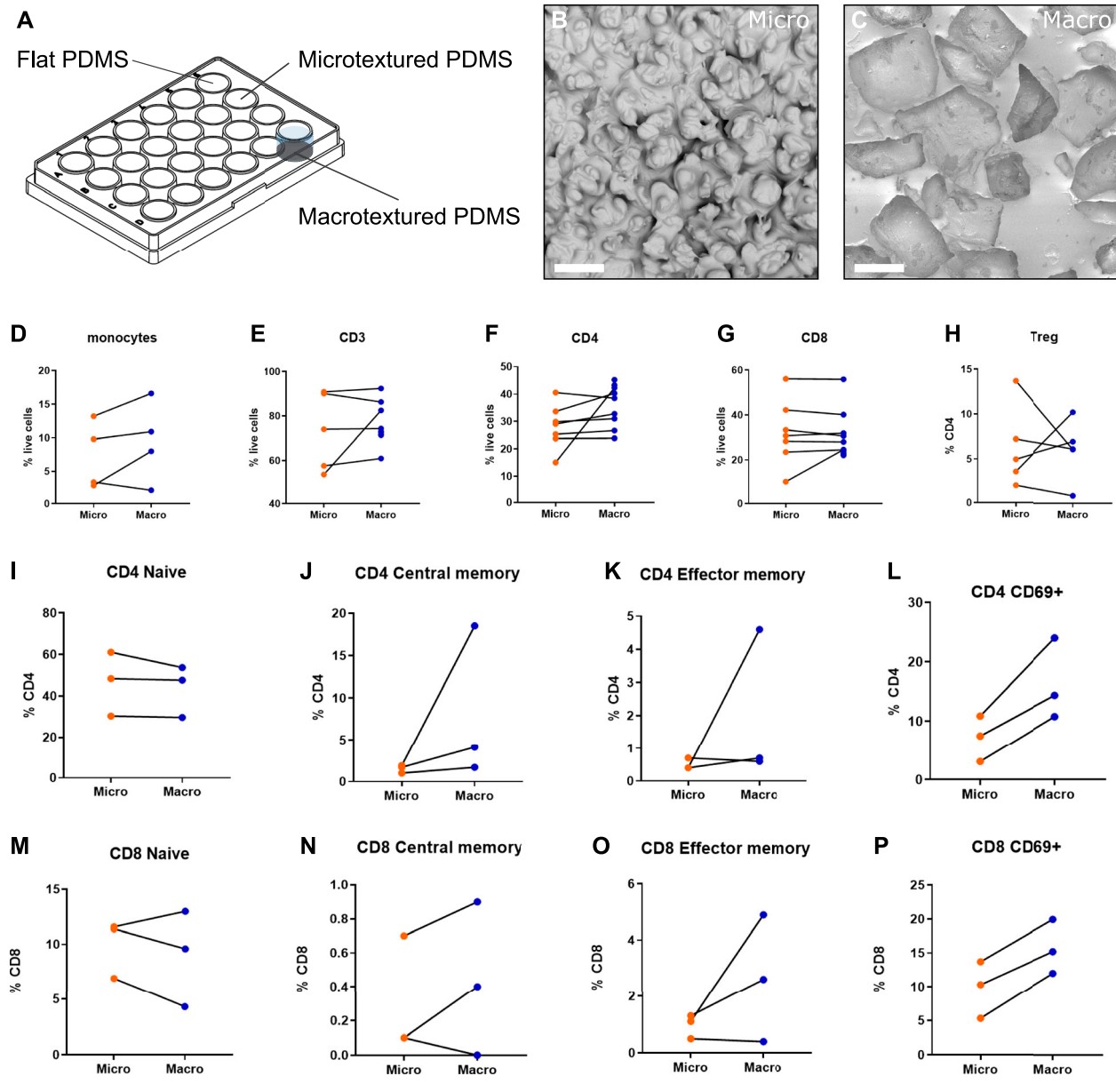

**Figure 2. Flow cytometry analysis of PBMCs cultured on model surface textures.**
**(A)** Schematic representation of the in vitro experimental model used to assess the immune response of PBMCs to different surface textures. **(B)** Scanning electron microscopy image of a PDMS replica of a microtextured surface (Mentor Siltex), showing rounded bumps and cavities, with dimensions ranging from 10 to 50 $\mu m$, created using the coating emulation technique. **(C)** Scanning electron microscopy image of a PDMS replica of a macrotextured surface (Allergan Biocell), showing cubic cavities characteristic of this texture, with dimensions ranging from 100 to 400 $\mu m$, obtained by the "salt loss" technique. **(D, E, F, G, H, I, J, K, L, M, N, O, P)** Results of flow cytometry analysis depicting various immune cell populations and their activation status in PBMCs cultured on these model surfaces.

backgrounds, the evidence suggests that genetic factors alone may not fully explain the observed differences in immune response. This inference is drawn from two key observations in our study. First, our in vitro results were derived from culturing cells from healthy donors on well plates with different surface textures. The observed differences in response across these textures hint at the influence of the implant's surface characteristics over a genetic predisposition in these controlled conditions. Second, in two unique cases, we analyzed cells from the periprosthetic fluid of patients with bilateral implants of different textures. The variance in response

between these two implants within the same patient again points toward the impact of the implant's texturization. Although these observations indicate that the texture of implants plays a significant role in leukocyte activation, we acknowledge that they do not entirely rule out the influence of genetic factors. Future studies involving a comprehensive genetic analysis could provide more definitive insights into the interplay between the genetic background and implant texture in modulating immune responses.

The findings from our investigation, including the low bacterial load and the nature of the identified species, challenge the

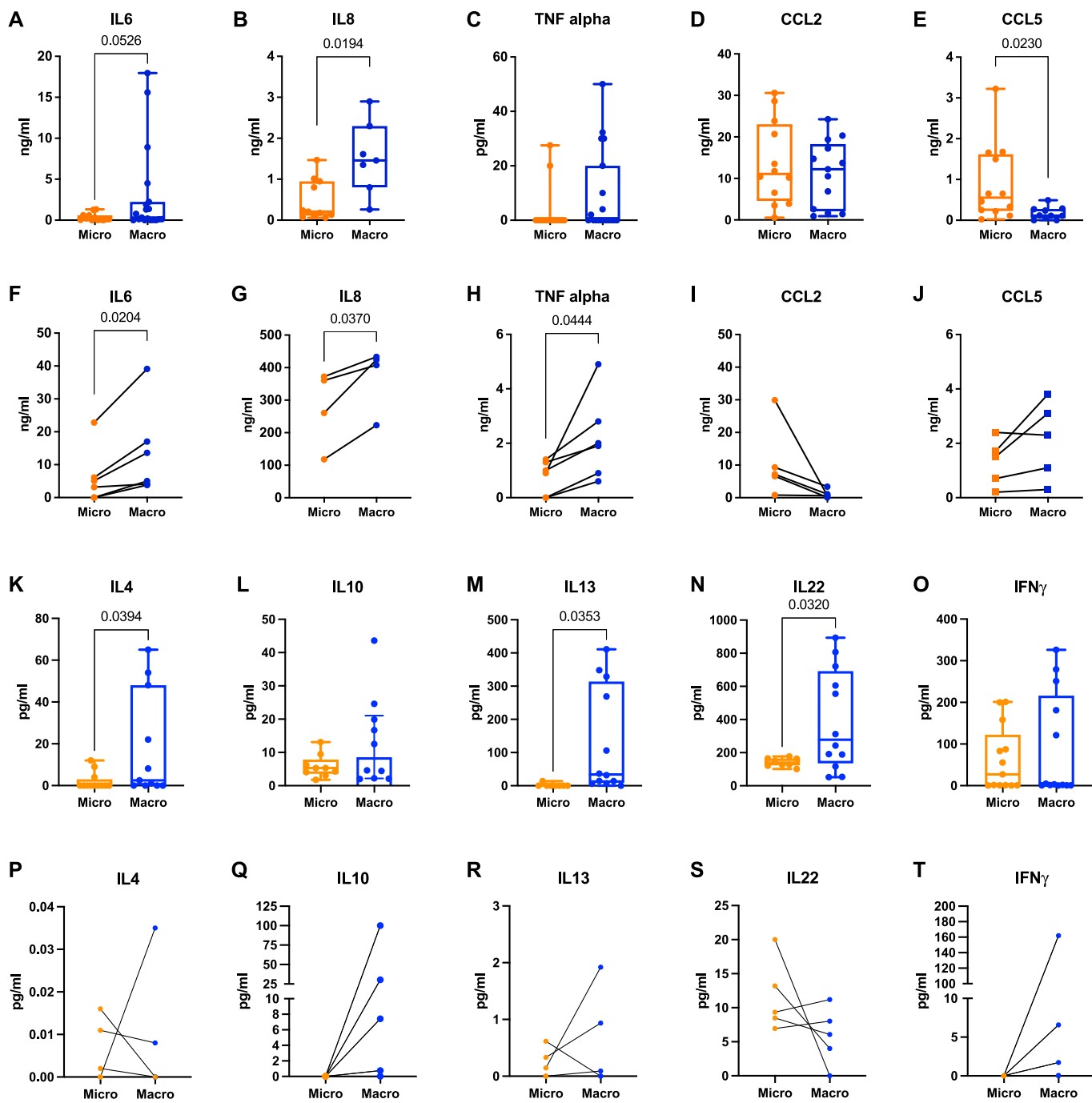

**Figure 3. Quantification of soluble mediators in periprosthetic fluids and supernatants of PBMCs cultured on microtextured or macrotextured surfaces.**
**(A, B, C, D, E)** ELISA quantification of IL6, IL8, TNF-alpha, CCL2, and CCL5 in the periprosthetic fluid of patients with either microtextured or macrotextured implants.
**(F, G, H, I, J)** ELISA quantification of the same set of cytokines and chemokines (as in (A, B, C, D, E)) released from PBMCs cultured on microtextured or macrotextured surfaces. **(K, L, M, N, O)** ELLA quantification of IL4, IL10, IL13, IL22, and INFɣ in the periprosthetic fluid of patients with microtextured and macrotextured implants.
**(P, Q, R, S, T)** ELLA quantification of the same set of cytokines (as in (F, G, H, I, J)) released from PBMCs cultured on microtextured or macrotextured surfaces. Histograms represent the mean ± SD, based on data from at least four independent donors. **(A, B, C, D, E, F, G, H, I, J, K, L, M, N, O, P, Q, R, S, T)** Statistical significance was determined using an unpaired *t* test with Welch's correction (A, B, C, D, E, K, L, M, N, O) and a paired *t* test (F, G, H, I, J, P, Q, R, S, T).

assumption that bacterial biofilms are the primary driver of the chronic inflammatory state linked to macrotextured breast implants. Notably, our experiments showed no significant difference in lymphocyte accumulation on macrotextured surfaces, both in the absence and presence of bacteria. These observations are particularly relevant considering the established correlation between biofilm formation and capsular contraction, which is thought to increase the risk of developing ALCL (Alessandri-Bonetti et al, 2023).

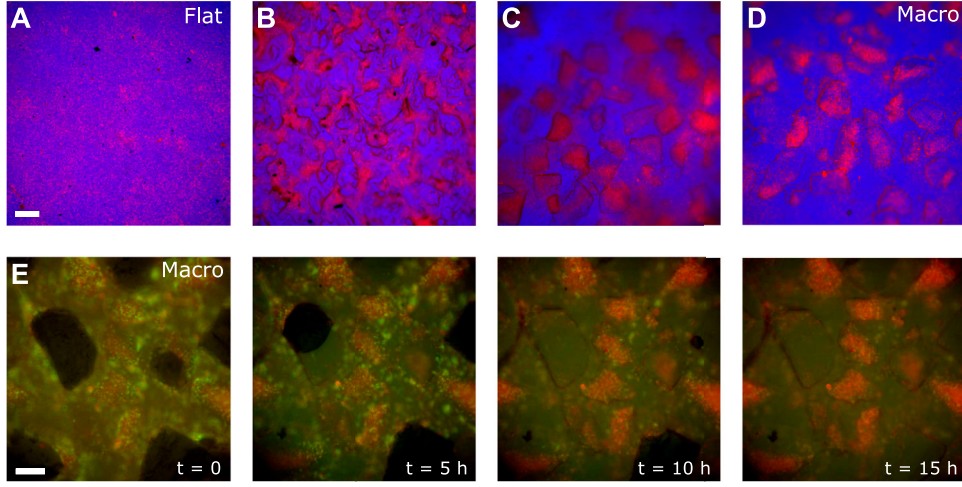

**Figure 4. Leukocyte distribution on model surface textures.**
**(A, B, C, D)** Overlay of phase-contrast (blue) and fluorescent images (red indicating PBMCs) acquired 12 h after plating cells on different surfaces: flat (A), microtextured (B), and macrotextured ((C) for donor 1, (D) for donor 2). Scale bar, 200 *μm*. **(E)** Overlay of phase-contrast and fluorescent images (green for monocytes, red for lymphocytes) acquired at intervals of 0, 5, 10, and 15 h from the beginning of the experiment. Here, cells were plated on a macrotextured surface and co-cultured with *S. epidermidis*. Scale bar, 100 *μm*.

Although the presence of bacteria has been a focal point in understanding the etiology of implant-associated complications, our results suggest that the physical properties of the implant surface itself may play a more pivotal role in initiating and sustaining chronic inflammatory responses.

Our study reveals that macrotextured surfaces are associated with elevated levels of pro-inflammatory cytokines IL6, IL8, and TNF-alpha in periprosthetic fluids from patients, pointing to an enhanced inflammatory response linked to the implants' specific topography. Furthermore, our cytokine profile analysis, focusing on markers like IL4, IL10, IL13, IL22, and INFɣ previously linked to BIA-ALCL, has identified increased levels in association with macrotextured surfaces. Moreover, we observed a heightened activation in both the CD8 and CD4 compartments on macrotextured surfaces, characterized by an increased presence of effector and central memory cells, alongside noteworthy elevations in Treg and CD69[+] cells, compared with microtextured ones. Particularly significant is the observed predominance of CD30[+] cells in the periprosthetic fluid of patients with macrotextured implants. Considering BIA-ALCL's hallmark association with CD30[+] markers (Quesada et al, 2019; Zhang et al, 2022), these results underscore a crucial link between implant surface texture and an immunological milieu that may predispose to lymphomas associated with implantable devices.

The observed higher prevalence of tumor-associated macrophages—typically marked by CD206 and CD163 and known for their immunosuppressive behavior (Belgiovine et al, 2020)—in the collected samples from patients with macrotextured implants further suggests that the polarization of macrophages may be influenced by the implant's surface topography. Such macrophages are implicated in promoting tumor development and survival and resistance to conventional antitumor treatments (Germano et al, 2011; De Palma & Lewis, 2013). In line with findings in BIA-ALCL tissues (Laurent et al, 2016), our study also indicates an increasing trend in NK cells, eosinophils, and neutrophils in the context of macrotextured prostheses. This points to a more pronounced inflammatory niche associated with these implants, potentially contributing to the chronic-like inflammatory environment observed. These comprehensive findings underscore the critical role of implant surface texture in shaping the immune response, possibly influencing the risk and progression of conditions such as BIA-ALCL.

The development of reliable and reproducible physical models for in vitro testing is crucial for conducting extensive, long-term experimental analyses of the biological effects of implanted prostheses. Such models offer an invaluable platform for investigating the direct mechanisms that may lead to possible adverse events. Factors such as surface area, roughness, the depth of cavities and pores, and the structured of edges have been previously indicated as significant contributors to the body's reactions to implants (Barr et al, 2017; Loch-Wilkinson et al, 2020; Belgiovine et al, 2023). Our in vitro cultures using PBMCs on the model surfaces created for this study not only demonstrated the biocompatibility of the material used (PDMS) but also provided insights into the cellular responses to different surface topographies. The cytokine release patterns observed in these in vitro experiments were consistent with those seen in ex vivo samples from patients. This consistency strengthens the relevance of our model for mechanistic studies, highlighting its capability to accurately replicate the biological interactions occurring in the body postimplantation.

We observed that leukocytes, particularly lymphocytes, tend to be captured within the cavities of macrotextured surfaces. This entrapment appears to stimulate these cells to release inflammatory cytokines, a finding in line with recent evidence linking lymphomas with implantable devices (Brody, 2016). In addition, the concept of "tribology"—the study of friction between interacting surfaces—is gaining attention in the context of implant carcinogenicity, as seen in orthopedic implants (Clemens et al, 2019). This aspect of physical interaction between implant surfaces and biological tissues offers a new perspective on the mechanisms underlying chronic inflammation. Previously, it was hypothesized that chronic inflammation might be linked to the aging and wear of silicone prostheses, suggesting that silicone itself could trigger an inflammatory reaction (Bizjak et al, 2015). However, our findings indicate that the topographical features of the implant surface, rather than the silicone material per se, might play a more significant role in initiating and sustaining inflammatory reactions.

The impact of surface topography on implant-associated inflammation is increasingly being recognized as a critical factor in the establishment of a pro-tumoral environment, potentially leading to conditions like BIA-ALCL. Consequently, the development of less irritating and more inert implant surfaces may hold promise for improving the safety and efficacy of prosthetic devices across various medical fields. Given the potential health implications, ongoing research in this area is paramount importance. Recent studies have expanded the correlation between lymphomas and implantable devices beyond breast prostheses. Lymphomas have been linked to a range of devices, including cardiac, joint, gluteal, testicular, and intraocular implants, made from both silicone and non-silicone materials (Moruzzo et al, 2009; Palraj et al, 2010; Sanchez-Gonzalez et al, 2013; Kellogg et al, 2014; Vivacqua et al, 2015). Although genetic predisposition plays a role, it is evident that implants themselves can influence the inflammatory response in the body. Moreover, concerns extend beyond lymphomas. Studies have shown associations between breast implants and autoimmune diseases and a heightened incidence of breast cancer recurrence in patients who underwent heterologous reconstruction postmastectomy (Watad et al, 2018; Lee et al, 2020; Tervaert et al, 2022). Our findings underscore the importance of surface texture in eliciting immune responses and suggest that the chronic inflammation observed may be more directly attributable to the physical characteristics of these implants than previously understood, potentially increasing the oncological risk for patients.

Although our study provides crucial insights into the immune responses triggered by different breast implant textures, we recognize the limitations inherent in our research, especially when considering a condition as rare as BIA-ALCL. This challenge is compounded by the difficulty in assembling a large, diverse sample pool, which impacts the statistical robustness and limits the scope of our conclusions. In our cohort of patients, we did not encounter any cases of BIA-ALCL. However, the observed increase in cytokines such as IL10 and IL13 in fluids from patients with macrotextured implants, and similar findings in our in vitro model, may hint at a potential dysregulation associated with these implant textures. Yet, we must exercise caution when extrapolating these results to the specific context of BIA-ALCL. Further research, involving a larger and more diverse sample population, is crucial to validate our findings within the unique pathology of BIA-ALCL. We also acknowledge the presence of potential biases in our study, including selection bias because of our choice of specific implant types and textures and observer bias in data interpretation. We have endeavored to mitigate these biases through careful experimental design, data collection, and analysis, yet complete elimination of these biases is challenging.

This research represents an initial exploration into a complex and evolving field. It sets the stage for future investigations, underscoring the need for more expansive and diverse studies to fully understand the interactions between breast implants and the immune system. Prospective studies, in particular, are essential to provide a deeper assessment of the long-term effects of implant surface textures on immune responses. Such studies hold the potential to guide the design of safer and more biocompatible prosthetic devices, ultimately improving patient outcomes. Our findings, therefore, not only contribute to the existing body of knowledge but also open avenues for further research that can significantly impact the field of implantable medical devices.

# Materials and Methods

### Patients' selection

We enrolled in the study 43 patients who had an intact breast implant (including both breast prostheses and expanders), with the criteria that these implants were neither exposed to the external environment nor infected. These encompassed cases of both breast prosthetic replacement (for esthetic or reconstructive reasons) and second-stage breast reconstruction. A total of 53 breasts were collected; of these, 24 breasts (45.3%) had macrotextured implants, whereas the remaining 29 breasts (54.7%) were fitted with microtextured devices. Notably, 34 of the 53 breast samples (64.2%) were collected from patients with a history of breast cancer. Radiotherapy was administered in 11 out of the 53 cases (20.7%). In terms of implant duration, only samples that had been in place for a minimum duration of 6 mo were included. In particular, 34 implants (64.2%) were in place for a period ranging from 6 to 24 mo, whereas the remaining 19 implants (35.8%) were retained for over 2 yr. All patients were provided with written informed consent which was signed before surgery. The study was approved by the Local Ethics Committee (reference number 163/21; CE Humanitas). Detailed clinical data of patients enrolled are reported in Table S1.

### Sample collection

In all enrolled subjects, the periprosthetic fluid was collected using a 10-ml sterile syringe immediately after making an incision of the capsule. The collected fluid was then promptly transferred to a sterile container for analysis. For shotgun metagenomic analysis, an aliquot of the collected periprosthetic fluid was stored at −80°C immediately after collection. For the evaluation of the immune landscape, collected periprosthetic fluid samples were centrifuged for 10 min at 780$g$ to separate cells from the liquid phase. Cells were analyzed by flow cytometry, whereas the liquid phase was used to measure the soluble molecules by ELISA and ELLA, as described below.

### Shotgun metagenomic analysis

Microbial DNA was extracted from 20 periprosthetic fluid samples using a commercial, ultrasensitive kit (Ultra-Deep Microbiome Prep; Molzym GmbH). However, seven of these samples were found to have DNA concentrations too low and were subsequently excluded from the shallow shotgun profiling that was performed by GenProbio. Out of 13 samples analyzed, 7 were associated with macrotextured implants and 6 with microtextured implants. This sample set included two patients with bilateral prostheses, each featuring different surface topographies.

### Ex vivo flow cytometry analysis

Flow cytometry was performed after standard procedures (Cossarizza et al, 2021) on cells derived from periprosthetic fluids collected from 16 patients with macrotextured implants and 27 patients with microtextured implants. FACS analysis was not carried out for all samples because some samples were inadequate, in terms of quantity or contamination. Samples were stained with a live-dead exclusion dye (LIVE/DEAD Fixable Aqua Dead Cell Stain Kit) for 15 min at RT to discriminate dead and viable cells. Subsequently, cells were incubated with the monoclonal antibodies listed in Table S2 for surface antigen staining. After the staining, labeled cells were fixed in PBS + 1% formalin. The acquisition was performed at FACSymphony A5 (BD Biosciences). FACS data were analyzed with FlowJo X 10.0.7r2 software (BD).

### In vitro flow cytometry analysis

We used human PBMCs isolated from the buffy coats of healthy female donors. The PBMCs were separated using a Histopaque-1077 gradient (Sigma-Aldrich) and then cultured on model surface with different textures, previously exposed to UV light for sterilization. PBMCs were plated at a concentration of $2 \times 10^6$ cells per well and cultured for 48 h at 37°C with 5% $CO_2$ in 24-well microplates (Costar; Corning Incorporated). After 48 h of culture on texture surface replicas, PBMCs were collected for FACS analysis. Cell viability was assessed using 7AAD staining (BD Biosciences), following the manufacturer's instructions. To assess identity immune cell populations and evaluate their activation state, cells were incubated with the monoclonal antibodies listed in Table S2 for surface antigen staining. After the staining, labeled cells were fixed in PBS + 1% formalin. The acquisition was performed at FACSymphony A5 (BD Biosciences). A minimum of 50,000 events were acquired for each sample. Data were analyzed using BD FACSDiva 8.0.1 software (BD Biosciences) and FlowJo X 10.0.7r2 software (BD).

### Model surface preparation

Model surfaces made of silicone elastomer were prepared to replicate microtextured and macrotextured implants. For macrotextured surfaces, we employed the "salt loss" technique, whereas a double replication process was used for microtextured surfaces. For this purpose, we selected SYLGARD 184 (Dow Corning), a low-viscosity, transparent PDMS polymer, chosen for its biocompatibility, non-cytotoxicity, and excellent flow properties (Rusconi et al, 2014). These characteristics are crucial for accurate surface replication and easy inspection of the final components. The viscosity of SYLGARD 184 was monitored during curing to evaluate the evolution of its flow properties and determine the optical gelation time. Replicas of microtextured and macrotextured surfaces were created by pouring and curing the elastomer in polystyrene wells under identical conditions as those employed for the textured models (Figs 2 and S2). In addition, to provide a baseline comparison, we fabricated non-textured control surfaces using the same silicone rubber.

### Diagnostic assays

The expression levels of the analyzed soluble molecules were investigated in both periprosthetic fluids and in culture supernatants by ELISA and ELLA. Periprosthetic samples were processed as previously described. Supernatant fluids were collected after 48 h of culture and centrifuged at 285g for 5 min before proceeding with the analysis. For the quantification of human cytokines IL-6, IL-8, TNF-alpha, CCL2, and CCL5, we used commercial ELISA kits (R&D Systems) according to the manufacturer's guidelines. The data obtained from these assays were subsequently analyzed using SoftMax Pro 5.3 software. The ELLA system (ProteinSimple; Bio-Techne) for automated enzyme-linked immunoassays of the cytokines IL4, IL6, IL8, IL10, IL13, IL22, and INFγ. ELLA, based on microfluidic technology, allows for the performance of these assays with minimal manual intervention. Samples were diluted 1:1 with the washing buffer and pipetted into the instrument's cartridge. Each cartridge comes with a pre-generated calibration curve by the manufacturer for each lot, and the ELLA system reads the cartridge's barcode to acquire calibration-related parameters. The quantification of cytokines was then performed based on these master calibration curves, with fluorescent signals being read and processed internally by the ELLA instrument.

### Microscopy experiments

PDMS surface replicas mimicking various textures, placed in the bottom of a polystyrene 24-well chamber, were first sterilized under UV light. PBMCs, pre-stained with CellTracker Fluorescent Probes (Thermo Fisher Scientific Inc.), were seeded onto these surfaces. In experiments involving bacterial interaction, a suspension of *S. epidermidis* (strain ATCC 12228, OD = 0.1) in Tryptone Broth was added to the wells and incubated for 24 h at 37°C, after which the bacterial medium was replaced with the PBMC solution. Imaging was performed on a DMI8 Leica microscope, utilizing a 20× air objective and maintained under climate control at 37°C. For each experimental condition, 10 images across different vertical planes were captured using an ORCA-Flash 4.0 V3 Digital CMOS camera (Hamamatsu). Time-lapse imaging was conducted at 10-min intervals for up to 5 h, employing Metamorph (v7.10.1.161) for image acquisition.

### Statistical analysis

Statistical analysis was conducted using an unpaired *t* test with Welch's correction or paired *t* test, as indicated (GraphPad 9, Prism statistical package). Continuous variables, encompassing the data derived from cytokine quantification and cell behavior analysis, are presented as mean ± SD. A *P*-value of less than 0.05 was set to determine statistical significance.

## Supplementary Information

# Acknowledgements

We acknowledge support from the Italian Ministry of University, Bando PRIN 2020, Grant 2020LYY27Z (to M Klinger, V Vinci, G Janszen, and L Di Landro), and from the Italian Ministry of Health, Fondi 5x1000 Ricerca Sanitaria (to R Rusconi).

## Author Contributions

V Vinci: conceptualization, resources, funding acquisition, investigation, and writing—original draft.
C Belgiovine: conceptualization, data curation, formal analysis, investigation, and writing—original draft.
G Janszen: conceptualization, data curation, supervision, investigation, methodology, and writing—original draft.
B Agnelli: data curation, formal analysis, investigation, and methodology.
L Pellegrino: data curation, formal analysis, investigation, and methodology.
F Calcaterra: data curation, formal analysis, investigation, methodology, and writing—review and editing.
A Cancellara: data curation, investigation, and methodology.
R Ciceri: data curation, investigation, and methodology.
A Benedetti: data curation, formal analysis, investigation, and methodology.
C Cardenas: data curation, formal analysis, investigation, and methodology.
F Colombo: data curation and methodology.
D Supino: data curation and methodology.
A Lozito: data curation and methodology.
E Caimi: data curation and methodology.
M Monari: data curation and methodology.
FM Klinger: data curation.
G Riccipetitoni: data curation and supervision.
A Raffaele: data curation and supervision.
P Comoli: data curation and supervision.
P Allavena: data curation and supervision.
D Mavilio: data curation, supervision, and investigation.
L Di Landro: conceptualization, data curation, supervision, investigation, methodology, and writing—original draft.
M Klinger: conceptualization, data curation, supervision, investigation, methodology, and writing—original draft.
R Rusconi: conceptualization, resources, data curation, formal analysis, supervision, funding acquisition, investigation, project administration, and writing—original draft, review, and editing.

## Conflict of Interest Statement

The authors declare that they have no conflict of interest.

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
