## [Reviewer comments · Life Science Alliance]

Life Science Alliance

Breast implant surface topography triggers a chronic-like inflammatory response

Valeriano Vinci, Cristina Belgiovine, Gerardus Janszen, Benedetta Agnelli, Luca Pellegrino, Francesca Calcaterra, Assunta Cancellara, Roberta Ciceri, Alessandra Benedetti, Cindy Cardenas, Federico Colombo, Domenico Supino, Alessia Lozito, Edoardo Caimi, Marta Monari, Francesco Klinger, Giovanna Riccipetioni, Alessandro Raffaele, Patrizia Comoli, Paola Allavena, Domenico Mavilio, Luca Di Landro, Marco Klinger, and Roberto Rusconi

DOI: <https://doi.org/10.26508/lsa.202302132>

Corresponding author(s): Roberto Rusconi, Humanitas University and Valeriano Vinci, Humanitas University

Review Timeline:

Submission Date:	2023-05-04
Editorial Decision:	2023-08-21
Revision Received:	2024-02-04
Editorial Decision:	2024-02-08
Revision Received:	2024-02-12
Accepted:	2024-02-13

Transaction Report:

August 21, 2023

Re: Life Science Alliance manuscript #LSA-2023-02132-T

Prof. Roberto Rusconi
Humanitas University
Biomedical Sciences
Via Rita Levi Montalcini 4
Pieve Emanuele
Italy

Dear Dr. Rusconi,

Thank you for submitting your manuscript entitled "Breast implant surface topography triggers an inflammatory response" to Life Science Alliance. The manuscript was assessed by expert reviewers, whose comments are appended to this letter. We invite you to submit a revised manuscript addressing the Reviewer comments.

Thank you for this interesting contribution to Life Science Alliance. We are looking forward to receiving your revised manuscript.

Sincerely,

B. MANUSCRIPT ORGANIZATION AND FORMATTING:

Reviewer #2 (Comments to the Authors (Required)):

The authors examine and compare the cellular composition and cytokines in macro- vs. micro-textured implants. Most implants were of relatively short duration compared to that required to develop BIA-ALCL. No patient studied had BIA-ALCL so the results are of doubtful significance to the pathogenesis of BIA-ALCL. In support of this, the principal cytokines found to be increased IL-6, IL-8 and TNF-a are not associated with published reports of cytokines in BIA-ALCL (IL-10, IL-9, IL-13) and there was no significant increase in CD30, the hallmark of ALCL.

Reviewer #4 (Comments to the Authors (Required)):

Summary of the paper:

This is an ex-vivo study of periprosthesis fluid derived from both micro and macro textured breast implants and proposed mechanism that may lead to breast implant associated anaplastic large cell lymphoma(BIA-ALCL).

Points of interest within the breast cancer clinical field are a proposed mechanism for the rationale of this pathology.

Overview of paper - the paper is particularly long, and descriptive in its results. The overall numbers are small.

It is interesting and there is detailed review in a simulated setting of what may be happening on a pathophysiological level. This may highlight areas for further study.

The layout is awkward in the full document, there is a lot of assay work and descriptive results with intermingled discussion and postulation. The material and methods should be presented prior to results.

Comments:

Abstract

"Here, by analyzing periprosthetic fluids from patients with macrot textured and micro textured breast implants and by cell culture experiments on different surface textures, we demonstrate that macrot textured surfaces trigger the activation of leucocytes in terms of inflammatory cytokines more than in the presence of micro textured surfaces"

This could be clarified - as this is specifically intraoperative harvest, as initially this could be aspiration of periprosthetic fluid. with ex vivo study.

Introduction

This sets the scene nicely, there is reasonable referencing, with international standards being referenced.

However there is lack of clarity regarding hypothesis or aims rather a description of what has been done - prior to starting the experiment what were the aims and objectives?

Lines 80-84 - can this be clarified and edited:

"Here, we analyzed the immune cells and inflammatory cytokines landscape present in the periprosthetic fluids of patients with different breast implants' surface textures. We found that patients with macro textured implants show an inflammatory microenvironment. In vitro experiments on model surfaces reproducing the micro and macro textured implants features that were observable ex vivo, supported these data"

Material and methods - this appears to be set at end which makes the paper difficult to interpret.

General comments -

The patients - were this benign replacements or cancer patients - this should be stated and described as the patient group.

There needs to be clarity regarding the ex-vivo "implants" are not actual implants but prepared model surfaces - whether this is truly representative of in-market implants is difficult to determine but should be considered a significant limitation is extrapolating this to the "real world" pathophysiology.

Small numbers - statistically analysis of FACS with a total 7 cases per limb ? value, and whether there was an attempt at powering this.

for the FACS 14 cases were used - these should be reported as breasts/samples rather than patients given that there are bilateral cases. I would be keen to know what the 22 sample excluded were excluded. - with reference to the table (supplementary table 1.

Ethics approved is stated. Conflicts of interests are also clear.

Results:

Line 49 the use of the word precocious - I'm not too sure this makes sense.

There are several areas in the results line 112-114 where the conclusion are intermingled with the results. This should be edited.

The time the

Overall the experiment is interesting as the results, but the generalisability is limited at the stage. Further prospective controlled worked would be helpful.

Harvest of insitu implants at a 5-10 year time span would also be more relevant for clinic ALCL.

Graphs are clear - some of the P-values likely are underpowered for significance.

Although the bacterial portion is mildly interesting this is not stated in the aims/or the methods as to this being part of the original study.

Discussion 191-194 - the exclusion of genetic component - this seems a little bit of a leap. - should be phrased in a less decisive manner.

line 213-214 "more present" grammatically a little awkward

In the discussion the limitation of tiny number particularly with the rare prevalence of BIA ALCL should be acknowledged.

Further acknowledgement of limitations, and potential bias would be helpful.

Reply to Reviewer #2 (Reviewer comments are included in italics)

The authors examine and compare the cellular composition and cytokines in macro- vs. micro-textured implants. Most implants were of relatively short duration compared to that required to develop BIA-ALCL.

We thank the Reviewer for their comments, which have prompted us to conduct additional experiments and analyses to further enhance the robustness of our study. We acknowledge the valid concern regarding the relatively short duration of implantation compared to the typical onset time of BIA-ALCL. In response to this concern, we have made significant improvements to the manuscript. The initial submission included a total of 36 samples, with only 8 exceeding 24 months of implantation. In this resubmission, we have expanded our sample pool by collecting an additional 17 samples (from 15 patients), resulting in a total of 53 samples. Notably, 12 of these samples are derived from implants with more than 8 years of permanence in the body. To provide transparency and address the duration concern, we have revised Table 1 in the supplementary material to include specific information on the time of permanence for each implant from which periprosthetic fluid was collected.

These additions aim to provide a more comprehensive understanding of the relationship between breast implant surface topography and immune response over extended periods. Additionally, we will discuss the potential implications of our findings in the context of the observed short-term effects and highlight the need for further research involving longer-term follow-up to better understand the relationship between implant surface topography and BIA-ALCL development.

No patient studied had BIA-ALCL so the results are of doubtful significance to the pathogenesis of BIA-ALCL. In support of this, the principal cytokines found to be increased IL-6, IL-8 and TNF- α are not associated with published reports of cytokines in BIA-ALCL (IL-10, IL-9, IL-13) and there was no significant increase in CD30, the hallmark of ALCL.

We thank the Reviewer for the opportunity to address concerns regarding the significance of our results to the pathogenesis of BIA-ALCL. We acknowledge that our patient cohort did not include individuals diagnosed with BIA-ALCL. However, it is crucial to emphasize that the primary objective of our study was to investigate the immune response elicited by different breast implant surface topographies. While our findings may not directly apply to BIA-ALCL, they offer valuable insights into the potential inflammatory effects of macrotextured implants, which could be relevant to the broader context of implant safety and associated lymphomas, as noted in the literature.

In response to the Reviewer's comment regarding cytokine profiles and CD30 expression, since we have expanded the sample pool for this resubmission, we are pleased to report that our additional analyses have yielded significant findings. Notably, we observed a statistically significant increase in CD30 expression ($p = 0.0202$), as detailed in the revised Figure 1, panel w. Additionally, our new flow cytometry results reveal a statistically significant increase in regulatory T lymphocytes (Fig. 1l) and CD69 positive cells (Fig. 1u) on macrotextured surfaces compared to microtextured ones. This suggests a shift towards immune regulation rather than aggression, a hypothesis further supported by the observed reduction in highly inflammatory monocytes (Fig.

1a) and an increase in macrophages, especially those with immunosuppressive characteristics (Fig. 1c and 1d). These results, together with the confirmation of previously observed trends, paint a comprehensive picture of a chronic-like immune response in the presence of macrotextured implants. While our study does not directly involve BIA-ALCL patients, these insights open intriguing possibilities for understanding the immune mechanisms potentially related to different implant surfaces. They may also provide valuable context for future research into the pathogenesis of BIA-ALCL and similar conditions.

Furthermore, following the Reviewer's suggestion, we have broadened our cytokine analysis, including IL10, IL13, IL4, IL22, and $\text{INF}\gamma$, all of which have been associated with ALCL in existing literature. For this expanded analysis, we utilized ELLA technology, offering higher sensitivity and the capability for simultaneous multiple immunoassays, an advantage over traditional ELISA methods. Our expanded experiments encompassed 27 samples, comprising 14 from our initial patient cohort and 13 new ones, detailed in the revised Supplementary Table 1. The new results revealed a statistically significant increase in the release of IL13, IL4, and IL22 for macrotextured surfaces compared to microtextured ones. These findings, corroborated by in-vitro results on replica surfaces for IL4, IL10, IL13, and $\text{INF}\gamma$, are now illustrated in the revised Figure 3 (panels k-o and p-t). Additionally, results obtained with ELLA are consistent with those from ELISA for IL6 and IL8, as demonstrated in the new Supplementary Figure 4.

These findings not only validate our initial observations but also significantly extend the understanding of the cytokine profile associated with different breast implant textures. We believe that this expanded analysis underscores the potential relevance of our research to the pathogenesis of BIA-ALCL and strengthens the relevance of our study to the broader discussion on breast implant-associated lymphomas.

Reply to Reviewer #4

Summary of the paper:

This is an ex-vivo study of periprosthesis fluid derived from both micro and macro textured breast implants and proposed mechanism that may lead to breast implant associated anaplastic large cell lymphoma(BIA-ALCL).

Points of interest within the breast cancer clinical field are a proposed mechanism for the rationale of this pathology.

We are grateful to the Reviewer for recognizing the importance of our study in advancing the understanding of BIA-ALCL within the clinical context of breast cancer. Our ex-vivo analysis of periprosthetic fluids from different breast implant textures, along with supportive in-vitro results, provides preliminary insights into the immune mechanisms potentially leading to BIA-ALCL. We believe this work lays a solid foundation for future research, potentially guiding clinical decisions and enhancing patient safety in breast implant surgeries.

Overview of paper - the paper is particularly long, and descriptive in its results. The overall numbers are small.

We acknowledge the Reviewer's concern regarding the length and descriptive nature of our manuscript. We have made efforts to ensure that the detailed results presented are essential to convey the complexity and significance of our findings in understanding the immune response to different breast implant textures.

Regarding the sample size, we are pleased to report that we have significantly expanded our cohort in the revised manuscript. We have now analyzed data from 53 samples, derived from 43 patients, including 10 with bilateral breast prostheses. This enlargement of our sample size has been particularly impactful in our flow cytometry (FACS) analyses, where we now observe a statistically significant difference in the increased expression of CD30, T-regulatory cells (T-reg), and CD69 in macrotextured surfaces compared to microtextured ones, as illustrated in the revised Figure 1. This increase in sample size has not only strengthened our findings but also enhanced the statistical robustness of our conclusions. Additionally, we have broadened our panel of cytokines analyzed. This expansion of our cytokine analysis further contributes to the comprehensive nature of our study, providing deeper insights into the immune responses associated with different breast implant surfaces.

It is interesting and there is detailed review in a simulated setting of what may be happening on a pathophysiological level. This may highlight areas if further study.

We are pleased that the simulated setting of our research, which aims to closely replicate the in-vivo environment, has been well-received. We agree that our findings do indeed highlight several areas for further study. Our work opens up possibilities for more in-depth in-vivo research to validate and expand upon these initial insights. Future studies could explore the long-term immune response to various implant textures and their direct correlation with the development of BIA-ALCL, potentially leading to safer breast implant designs and improved patient outcomes.

The layout is awkward in the full document, there is a lot of assay work and descriptive results with intermingled discussion and postulation. The material and methods should be presented prior to results.

We thank the Reviewer for their feedback regarding the layout and organization of our manuscript. In response to this comment, we have carefully revised the structure of the paper to enhance clarity and we have relocated some portions of the results discussion to the dedicated discussion section at the end of the manuscript.

Regarding the placement of the Materials and Methods section, we would like to clarify that our decision to present it after the Results and Discussion sections is in accordance with the stylistic guidelines of the journal to which we are submitting (*Life Science Alliance*). We believe this format aligns with the journal's preferred structure and aids in maintaining consistency with other publications in this journal.

Comments:

Abstract

"Here, by analyzing periprosthetic fluids from patients with macrot textured and micro textured breast implants and by cell culture experiments on different surface textures, we demonstrate that macrot textured surfaces trigger the activation of leucocytes in terms of inflammatory cytokines more than in the presence of micro textured surfaces"

This could be clarified - as this is specifically intraoperative harvest, as initially this could be aspiration of periprosthetic fluid. with ex vivo study.

In the revised abstract, we have made it clear that the periprosthetic fluids were harvested intraoperatively, rather than being aspirated. We have also emphasized that our analysis, including both the examination of periprosthetic fluids and cell culture experiments on different surface textures, was conducted ex-vivo.

The revised sentence in the abstract now reads: "This study investigates the immune response elicited by different prosthetic surfaces, focusing on the comparison between macrot textured and micro textured breast implants. Through the analysis of intraoperatively harvested periprosthetic fluids and cell culture experiments on surface replicas, we demonstrate that macrot textured surfaces elicit a more pronounced chronic-like activation of leucocytes, as well as an increased release of inflammatory cytokines, in contrast to micro textured surfaces".

Introduction

This set the scene nicely, there are reasonable referencing, with international standards being referenced. However there is lack of clarity regarding hypothesis or aims rather a description of what has been done - prior to starting the experiment what were the aims and objective?

We thank the Reviewer for the positive remarks about the initial setup and referencing in our Introduction. We appreciate the observation regarding the need for greater clarity in presenting our research aims and hypothesis. In response to this comment, we have revised the Introduction to more distinctly articulate the aims and objectives of our study. We explicitly state that our primary aim was to investigate the differential immune response triggered by macrot textured and micro textured breast implant surfaces. Additionally, we hypothesized that macrot textured surfaces would induce a more robust inflammatory response compared to micro textured surfaces, potentially contributing to the pathogenesis of conditions like BIA-ALCL.

Lines 80-84 - can this be clarified and edited:

"Here, we analyzed the immune cells and inflammatory cytokines landscape present in the periprosthetic fluids of patients with different breast implants' surface textures. We found that patients with macro textured implants show an inflammatory microenvironment. In vitro experiments on model surfaces reproducing the micro and macro84 textured implants features that were observable ex vivo, supported these data"

In the revised manuscript, we have rephrased this section which now reads as follows:

“In this study, our objective was to investigate the differential microbial contamination and immune responses elicited by macrot textured and micro textured breast implant surfaces. We hypothesized that macrot textured surfaces would provoke a more pronounced inflammatory response than micro textured surfaces, potentially playing a role in the pathogenesis of conditions such as BIA-ALCL. To explore this hypothesis, we conducted an extensive analysis of periprosthetic fluids collected from patients implanted with breast implants having different surface textures. Our approach encompassed examining bacterial load, profiling immune cell populations, and analyzing the inflammatory cytokine landscape. Our findings indicated a chronic-like inflammatory environment associated with macrot textured implants. Furthermore, in vitro experiments performed culturing healthy donor-derived peripheral blood mononuclear cells (PBMCs) on model surfaces mimicking the characteristics of both micro textured and macrot textured implants corroborated our initial observations, reinforcing the conclusion that the texture of breast implant surfaces is a critical factor in modulating the immune response in the periprosthetic environment”.

Material and methods - this appears to be set at end which makes the paper difficult to interpret.

We understand that the traditional structure of scientific papers places this section before the Results and Discussion sections, and we recognize that deviating from this format might pose challenges for some readers in terms of interpretability. However, we would like to clarify that the positioning of the Materials and Methods section at the end of the paper is in alignment with the stylistic guidelines of the journal (*Life Science Alliance*). That said, we are committed to ensuring that our paper is as accessible as possible. To this end, we have taken steps to ensure that the Materials and Methods section is clearly referenced throughout the paper, allowing readers to easily locate detailed methodological information when needed.

General comments -

The patients - were this benign replacements or cancer patients - this should be stated and described as the patient group.

In response to this comment, we have made significant enhancements to the manuscript to provide a clearer and more detailed description of the patients involved in our study. In our expanded cohort of 54 samples, 35 were obtained from breast cancer patients, 11 of whom had undergone radiotherapy. This detailed information, including the breast cancer classification and the time elapsed since the last radiotherapy treatment (ranging from 10 to 180 months), is now clearly stated in the Materials and Methods section and comprehensively presented in the revised Supplementary Table 1. It is important to note that, in our analysis, we did not stratify the results from flow cytometry (FACS) or the ELLA based on the patients' history of cancer or radiotherapy. This approach was taken to focus on the broader implications of implant surface textures on the immune response, irrespective of the patients' oncological history. By including these additional details in the revised manuscript, we aim to provide readers with a comprehensive understanding of our patient cohort.

There needs to be clarity regarding the ex-vivo "implants" are not actual implants but prepared model surfaces - whether this is truly representative of in-market implants is difficult to determine but should be considered a significant limitation is extrapolating this to the "real world" pathophysiology.

We acknowledge and appreciate the opportunity to clarify the distinction between our prepared model surfaces and actual in-market breast implants. In our research, we utilized model surfaces designed to closely mimic the topographical features of macrotextured and microtextured breast implants. These models were carefully crafted, drawing upon existing literature and implant specifications, to replicate the surface characteristics most relevant to the immune response. It is important to highlight that the fidelity of our models to in-market macrotextured and microtextured surfaces has been further validated by the approval of a patent in Italy, which is currently pending approval in the EU market. This patent confirms our ability to closely replicate the specific topographies of these implants, underscoring the relevance of our models to actual clinical scenarios. However, we recognize that despite this validation, our model surfaces are not exact replicas of specific in-market implants. This limitation is now explicitly addressed in the discussion section of our revised manuscript. We emphasize that while our models offer valuable insights into the immune responses triggered by different surface topographies, they may not fully capture the complete range of complexities and variables present in real-world clinical scenarios with in-market implants

Small numbers - statistically analysis of FACS with a total 7 cases per limb ? value, and whether there was an attempt at powering this. for the FACS 14 cases were used - these should be reported as breasts/samples rather than patients given that there are bilateral cases. I would be keen to know what the 22 sample excluded were excluded. - with reference to the table (supplementary table 1.

We understand the Reviewer's concerns about the robustness of our findings given the initial sample size and have taken steps to address this in our revised manuscript. In the original submission, we indeed faced limitations in the volume of periprosthetic fluid samples, which restricted the number of samples that could be analyzed through flow cytometry. This led to the use of only a subset of the available samples for this analysis. We acknowledge that this limitation was not clearly communicated in the manuscript. Indeed, 12 samples reported as unused in Table 1 were actually utilized, but not all markers were found in these samples, which might have led to some confusion.

To enhance the robustness of our study and address these issues, we have expanded our sample pool in the revised manuscript. We now include 18 additional samples, bringing the total to 43 samples analyzed through FACS. This larger sample size not only provides a stronger basis for our statistical analysis but also allows for a more comprehensive investigation of the immune response to different implant surfaces. Additionally, we have included an explanation in the revised manuscript for the 22 samples that were excluded, as detailed in Supplementary Table 1. These exclusions were primarily due to factors such as insufficient sample volume or compromised sample integrity, which we now explicitly state to provide full transparency in our methodology.

Ethics approved is stated. Conflicts of interests are also clear.

We appreciate the Reviewer's acknowledgment of the ethics approval and conflict of interest disclosures in our manuscript.

Results:

Line 49 the use of the word precocious - I'm not too sure this makes sense.

We thank the Reviewer for pointing this out. Upon reflection, we agree that the term may not convey our intended meaning clearly in this context. In response, we have taken the opportunity to reorganize the introduction of our revised manuscript. This restructuring was aimed not only at addressing this specific concern but also at improving the overall clarity and flow of the introduction. As part of this revision, the sentence in question has been removed. We believe these changes result in a more coherent and focused introduction, better aligning with our aims and hypotheses.

There are several areas in the results line 112-114 where the conclusion are intermingled with the results. This should be edited.

We thank the Reviewer for pointing this out. We have revised the lines in question, separating results from conclusions:

(Lines 82-92) "Periprosthetic fluids collected during the removal of both microtextured and macrotextured breast implants were first analyzed for bacterial contamination through a shotgun metagenomic approach (Supplementary Figure 1; Materials and Methods). The overall bacterial load in the periprosthetic fluids of our patient pool was generally low. However, the bacterial families identified, which included Actinobacteria, Bacteroides, Firmicutes, and Proteobacteria, demonstrated a similar richness of species across both types of implant surfaces (Supplementary Figure 1a-b). Predominantly, species from the Bifidobacterium genus were identified, which are also found in breast milk (Yan et al., 2021). This may suggest a potential link between the mammary glands and the prosthesis pocket. Notably, in two patients having bilateral prostheses with different surface topographies, a reduction in the number of bacterial species was observed in the macrotextured implant compared to the microtextured one (Supplementary Figure 1c-d)".

(Lines 212-221) "The findings from our investigation, including the low bacterial load and the nature of the identified species, challenge the assumption that bacterial biofilms are the primary driver of the chronic inflammatory state linked to macrotextured breast implants. Notably, our experiments showed no significant difference in lymphocyte accumulation on macrotextured surfaces, both in the absence and presence of bacteria. These observations are particularly relevant considering the established correlation between biofilm formation and capsular contraction, which is thought to increase the risk of developing ALCL (Alessandri-Bonetti et al., 2023). While the presence of bacteria has been a focal point in understanding the etiology of implant-associated complications, our results suggest that the physical properties of the implant surface itself may play a more pivotal role in initiating and sustaining chronic inflammatory responses."

Overall the experiment is interesting as the results, but the generalisability is limited at the stage. Further prospective controlled worked would be helpful.

We appreciate the Reviewer's recognition of the interest and value of our experiment and results. We understand that while our study provides important preliminary insights, its scope and design have certain inherent limitations. In the current study, due to the constraints related to the specific patient samples available, we recognize that our findings may not be broadly generalizable. However, we believe that our results provide a valuable foundation for further research in this area. This is now explicitly expressed in the discussion section of the revised manuscript (lines 292-305).

Harvest of insitu implants at a 5-10 year time span would also be more relevant for clinic ALCL.

In the revised manuscript we have expanded our sample pool, which now include 15 samples that have been in situ for more than 72 months, as detailed in the revised Supplementary Table 1.

Graphs are clear - some of the P-values likely are underpowered for significance.

In our initial study, the sample size and scope of experiments were indeed limited, which may have impacted the statistical power of our findings. Recognizing this limitation, we have expanded our sample pool and conducted additional FACS experiments to strengthen our statistical analyses. This expansion includes the addition of new samples, as mentioned earlier in our responses. With a larger dataset, we have been able to reassess our findings and achieve more robust statistical significance in key areas of our study, including the data on CD30 expression. The results from these additional experiments now demonstrate statistical significance with greater confidence.

Although the bacterial portion is mildly interesting this is not stated in the aims/or the methods as to this being part of the original study.

We acknowledge that the bacterial component, while a valuable aspect of our research, was not explicitly stated in the original aims or methods section of the manuscript. This oversight may have caused some confusion about its relevance and integration into the overall study. We have revised the manuscript to clearly articulate the role and significance of the bacterial analysis within the context of our research objectives. This amendment clarifies that the bacterial analysis was an integral part of our investigation into the immune response in the periprosthetic environment, aimed at providing a comprehensive understanding of the factors that might influence this response. We believe that the inclusion of this analysis enriches our study by offering additional insights into the microbial aspects of the periprosthetic space, which could have implications for understanding the immune response to breast implants.

Discussion 191-194 - the exclusion of genetic component - this seems a little bit of a leap. - should be phrased in a less decisive manner.

In the revised manuscript we have reframed the original statement to be less conclusive about excluding genetic factors, suggesting that while our observations indicate a significant role for

implant texture, they do not completely eliminate the possibility of genetic influence, leaving room for further investigation (lines 197-210).

line 213-214 "more present" grammatically a little awkward

We have replaced “more present” with “a higher prevalence”. The lines in question now read as follows (lines 236-239): “The observed higher prevalence of tumor-associated macrophages (TAMs) – typically marked by CD206 and CD163 and known for their immunosuppressive behavior (Belgiovine et al., 2020) – in the collected samples from patients with macrot textured implants further suggests that the polarization of macrophages may be influenced by the implant’s surface topography.”

In the discussion the limitation of tiny number particularly with the rare prevalence of BIA ALCL should be acknowledged.

We thank the Reviewer for highlighting the need to address the limitations of our study, especially concerning the small sample size and the rare prevalence of BIA-ALCL. We agree that these are significant points that warrant explicit acknowledgment in our discussion. In the revised version of our manuscript, we have added a section in the discussion explicitly addressing these limitations (lines 292-305). We recognize that the small number of samples, especially in the context of a condition as rare as BIA-ALCL, may limit the generalizability of our findings. This rarity poses a challenge in gathering a large and diverse sample pool, which in turn impacts the statistical power and the breadth of conclusions that can be drawn from our study. Furthermore, we discuss how the limited prevalence of BIA-ALCL makes it challenging to directly correlate our findings with the pathogenesis of this specific type of lymphoma. We emphasize that while our study offers important insights into the immune responses to different breast implant textures, caution must be exercised in extrapolating these results to the specific context of BIA-ALCL without further, more extensive research.

Further acknowledgement of limitations, and potential bias would be helpful.

In the revised manuscript, we have expanded the discussion section to provide a more comprehensive acknowledgment of the study’s limitations. We recognize that, in addition to the small sample size and the rarity of BIA-ALCL, there are other limitations that must be considered. These include the retrospective nature of the study, potential selection bias due to the specific patient population included, and limitations inherent in the methodologies and techniques used. We also acknowledge potential biases that may have influenced our results. This includes observer bias in data interpretation and potential biases introduced by the selection of specific surface textures and implant types for analysis. We emphasize that while we have made every effort to ensure the accuracy and reliability of our findings, these limitations and potential biases must be taken into account when interpreting the results.

February 8, 2024

RE: Life Science Alliance Manuscript #LSA-2023-02132-TR

Prof. Roberto Rusconi
Humanitas University
Biomedical Sciences
Via Rita Levi Montalcini 4
Pieve Emanuele 20072
Italy

Dear Dr. Rusconi,

Thank you for submitting your revised manuscript entitled "Breast implant surface topography triggers a chronic-like inflammatory response". We would be happy to publish your paper in Life Science Alliance pending final revisions necessary to meet our formatting guidelines.

- please be sure that the authorship listing and order is correct
- please upload all figure files as individual ones, including the supplementary figure files; all figure legends should only appear in the main manuscript file
- please add ORCID ID for the secondary and third corresponding authors -- they should have received instructions on how to do so
- please add the Twitter handle of your host institute/organization as well as your own or/and one of the authors in our system
- please make sure the author order in your manuscript and our system match
- please use the [10 author names et al.] format in your references (i.e., limit the author names to the first 10)
- please add your main, supplementary figure, and table legends to the main manuscript text after the references section
- please upload your Tables in editable .doc or Excel format
- we encourage you to revise the figure legend for Figure 1 such that all figure panels are introduced
- please add callouts for Figure S2A-D and Movie 1 to your main manuscript text

A. FINAL FILES:

B. MANUSCRIPT ORGANIZATION AND FORMATTING:

Sincerely,

February 13, 2024

RE: Life Science Alliance Manuscript #LSA-2023-02132-TRR

Prof. Roberto Rusconi
Humanitas University
Biomedical Sciences
Via Rita Levi Montalcini 4
Pieve Emanuele 20072
Italy

Dear Dr. Rusconi,

Thank you for submitting your Research Article entitled "Breast implant surface topography triggers a chronic-like inflammatory response". It is a pleasure to let you know that your manuscript is now accepted for publication in Life Science Alliance. Congratulations on this interesting work.

DISTRIBUTION OF MATERIALS:

Again, congratulations on a very nice paper. I hope you found the review process to be constructive and are pleased with how the manuscript was handled editorially. We look forward to future exciting submissions from your lab.

Sincerely,
